# Putting Laccase Gene Differences on Genomic Level into Context: An Analysis of *Botrytis cinerea* Strains from Grapes

**DOI:** 10.3390/microorganisms13030483

**Published:** 2025-02-21

**Authors:** Louis Backmann, Kim Marie Umberath, Pascal Wegmann-Herr, Fabian Weber, Andreas Jürgens, Maren Scharfenberger-Schmeer

**Affiliations:** 1Department of Biology, Chemical Plant Ecology, Technische Universität Darmstadt, Schnittspahnstrasse 4, D-64287 Darmstadt, Germany; louis.backmann@dlr.rlp.de (L.B.); juergens@bio.tu-darmstadt.de (A.J.); 2Institute for Viticulture and Oenology, Dienstleistungszentrum Ländlicher Raum (DLR) Rheinpfalz, Breitenweg 71, D-67435 Neustadt, Germany; pascal.wegmann-herr@dlr.rlp.de; 3Institute of Nutritional and Food Sciences, Molecular Food Technology, Agricultural Faculty, University of Bonn, Friedrich-Hirzebruch-Allee 7, D-53115 Bonn, Germany; umberath@uni-bonn.de; 4Section of Organic Food Quality, University of Kassel, Nordbahnhofstr. 1a, D-37213 Witzenhausen, Germany; fabian.weber@uni-kassel.de; 5Weincampus Neustadt, Hochschule Kaiserslautern, Breitenweg 71, D-67435 Neustadt, Germany

**Keywords:** strain comparison, *Botrytis cinerea*, laccase activity, laccase genes

## Abstract

One of the most important crop pathogens is *Botrytis cinerea*. It overcomes plant defenses using laccase, an enzyme which is frequently researched. Yet the differences between strains regarding their laccase activity is poorly understood. The aim of this study was to analyze laccase genes in the context of the regionality, vintage, and laccase activity of the strains. Eight strains were analyzed using whole genome sequencing, and the laccase activity was assessed. The strains were differentiated by SSR-PCR. We looked at all 14 known laccase genome regions as well as the promoter and terminator regions using variant metrics and phylogenetic trees. The laccase genes seem to be correlated with the regionality of the strains rather than the laccase activity, which provides new understanding to the study of pathogen adaption in specific environments. Some of the laccase gene regions showed little to no evolutionary change, while other regions showed a great variety of changes. This research highlights taking different laccase gene regions into context. We provide fundamental information for further research. Further studies, especially on gene expression, could provide insightful information regarding the potential of pathogen infection.

## 1. Introduction

Wine and grape production are of significant importance to the world, forming an integral part of historical and cultural development in many countries and having a high economic value [1,2,3,4]. As with many other agricultural sectors, the wine industry is facing a number of risks associated with the infection of crops during the maturation of grapes. One of the most significant pathogens affecting grapevines and grape berries is *Botrytis cinerea*, commonly known as grey mold. It is a necrotrophic fungal pathogen [5] that causes substantial yield and quality losses annually [4,6,7] even after the harvest of crops [1,8,9]. It is the second most prevalent pathogen, as indicated by [10,11], infecting over 200 relevant crop varieties and more than 1600 families of plants [2,12], predominantly in cold and humid climates [13], though it is also present in temperate and subtropical climates [11,13]. The pathogen facilitates multiple attack pathways and has the ability to infect the leaves, stems, fruits, and buds of the grapevine [8]. The fungus can enter grape parts at an early stage of ripening, remaining symptomless until a reactivation occurs later during harvest [14]. Alternatively, it can enter later during grape development through damaged tissue [15]. As the grapes continue maturing, the necrotrophic abilities result in rapid tissue decay [2]. The high virulence factors of *Botrytis* are primarily regulated by the production and expression of laccase [16,17]. In turn, the plant regulates defense through numerous strategies, for example by phytoalexines like resveratrol. Resistant grape varieties generally produce higher amounts of phytoalexines [18]. The fungus breaks down resveratrol into compounds more toxic for the pathogen [18], but some studies also suggested that higher laccase activity detoxifies trans-resveratrol [19,20], therefore providing better survivability of the pathogen. During infection, the penetration of the host tissue by *Botrytis cinerea* triggers a programmed cell death. The fungus then utilizes laccase as a multi-weapon tool, for example to detoxify plant defense mechanisms such as phytoalexines [18,21] or to oxidize a wide spectrum of polyphenolics [18]. Furthermore, a number of glycosylated laccase isoenzymes have been identified in different strains, with varying molecular masses, which suggests the involvement of different target enzymes in infection processes [22]. This renders the disease particularly challenging to control and to identify solutions for combating *Botrytis* on grapes. A variety of approaches may be employed to manage and control the disease. The most prevalent method thus far remains chemical control, as evidenced by numerous studies [23,24,25,26]. This is due to the fact that environmentally friendly alternatives, such as RNA interference, biological control, and plant resistance inducers, have not yet demonstrated comparable efficacy [24,27,28,29,30]. However, the use of chemical control measures against pathogens such as *Botrytis cinerea* may result in adverse environmental consequences, the development of resistant strains [24,31], and potential health concerns [32,33]. Furthermore, regulatory bodies such as the European commission are implementing measures to reduce the use of chemicals due to their adverse effects on the environment and human health [34,35,36]. A further consequence of climate change is the potential development of adapted strains. The likelihood of *Botrytis* developing more aggressive ways of infecting grapes is increased due to the prevalence of extreme weather conditions. In light of the emergence of rapidly adapting and resistant strains, a genomic approach is essential for a comprehensive understanding of pathogen evolution. The complete genome of *Botrytis cinerea* B05.10 has been sequenced [5,37]. Some studies focus on analyzing *Botrytis cinerea* in grapes at the genomic level with a view to identifying the key factors involved in infection [25]. Other studies investigate laccase genes [25] for the purpose of discriminating between botrytized and non-botrytized wines [38] or for transcriptomic analyses [39,40]. However, little is known about genomic differences between *Botrytis cinerea* strains. A study that employed resequencing to examine different *Botrytis cinerea* isolates from different geographical regions was conducted by [41]. The study’s objective was to gain a comprehensive overview and to investigate spontaneous mutation. With regard to their relationship with laccase activity, different regions and vintages have, to the best of our knowledge, not been placed into context. Furthermore, the majority of laccase genes have not been contextualized in relation to laccase activity across a range of different strains. In previous studies, we found that laccase activity had different impacts on phenolic compounds [42]. Understanding the impact of laccase genes on laccase activity can serve as an important anchor point [40]. It is therefore useful to understand the involvement of other laccase gene sites. These findings can serve as important tools to facilitate understanding and preparation for forthcoming changes in strain culture as well as for the development of new strategies in viticulture and oenology. Investigating the laccase gene sites can provide important points of departure for future analysis, such as gene expression studies. Analyzing gene sites benefits from the ability to distinguish different strains, finding potential markers for certain traits and bioinformatic tools. A quick and easy way to distinguish *Botrytis cinerea* strains is PCR, specifically SSR-PCR, as it provides a simple way to analyze the pathogen [43,44]. Using next generation sequencing (NGS) provides deep insights into the genome, facilitating high-throughput whole genome sequencing even with small quantities of starting material [45]. Classic evaluation tools used to highlight potential changes between species are phylogenetic trees, as they show evolutionary consensus between strains. Phylogenetic trees have been part of many research studies revolving around genomic analysis, specifically evolutionary and, in turn, environmental relationships [46,47].

This research analyses different *Botrytis cinerea* strains at the genomic level to identify potential differences between different *Botrytis* strains regarding region, vintage, grape variety, and laccase activity. We highlight strain differences in specific regions that may be responsible for infection efficiency. We hypothesize that laccase activity is recognizable at the genomic level, either by SNPs or InDels in the open reading frame or by alteration in the regulation of the laccase genes. Our aim is to provide an overview focused on laccase regions and laccase genes of *Botrytis cinerea*.

## 2. Materials
and Methods

### 2.1. Sampling and Strain Differentiation

The strains were collected during the 2020, 2021, and 2022 vintage, with an additional sampling conducted during the 2017 vintage. Satellite images of the collection are provided in the Appendix A. All strains were isolated from visible sporulating grape berries Figure 1 and subsequently transferred to malt agarose plates. In addition, previously collected strains from the Phytomedizin Institute (Dienstleistungszentrum Rheinpfalz, DLR) strain collection as well as the reference strain Bc05.10 were utilized. All strains were isolated from single spore isolates. Every strain was kept in duplicate during the experiments; the reference strain Bc05.10 (DSMZ strain) served as a control.

The selected regions were Bonn, Bordeaux, Diedesfeld, Heppenheim, and Edenkoben based on the availability of regions’ vintages and laccase activity. The collection site of Bonn is located in the mid-west of Germany in a urban region. The strain derives from a enclosed vineyard used for experiments. During the summer, more heat can buildup in the surrounding city, potentially influencing the grapes. There are no other vineyards around. The Bordeuax strain derives from the suburban Barsac region of Bordeaux with many vineyards around. The typical weather can favor the production of sweet wines with early morning fog or high humidity and drier middays. Typical weather is warm during harvest season. The vineyards are more prone to the environment and surrounded by other vineyards. The Diedesfeld and Edenkoben collection sites are in close proximity to each other and are both from a suburban region in the west of Germany. Typical weather is more cold and wet and reaches moderate temperatures of up to 30 °C with a few exceptions during the summer where it can get warmer. Both collection sites are plain and open to the environment with other vineyards surrounding them. The collection site of Heppenheim is located roughly 70 km apart from Diedesfeld and Edenkoben on a small mountain surrounded by other vineyards. Typical weather is similar to Diedesfeld and Edenkoben, although wind properties are different, and the temperature is a little bit lower due to the higher elevation. *Botrytis cinerea* strains were cultivated towards sporulation, and DNA was extracted using the RED Extract Plant PCR Kit (Merck KGaA, Darmstadt, Germany). The strains were differentiated by SSR-PCR as previously described [48]. The procedure was as follows: For DNA extraction, the samples were grown for two weeks towards sporulation. A small sample amount of 100 mg was scratched from the agar plates using forceps. The RED Extract Plant PCR kit from Merck KGaA, Darmstadt, Germany was used for extraction. Next, 100 µL of extraction solution was added to the sample and incubated at 95 °C for 10 min. Directly after, 100 µL of dilution solution was added to the sample and vortexed to mix. The solution was centrifuged for 10 min at 13,000× *g*, and the soluble phase was kept for further experiments. The SSR-PCR was conducted using eight simple sequence repeat markers [43]; Bc1F: AGGGAGGGTATGAGTGTGTA, Bc1R: TTGAGGAGGTGGAAGTTGTA, Bc2F: CATACACGTATTTCTTCCAA, Bc2R: TTTACGAGTGTTTTTGTTAG, Bc3F: GGATGAATCAGTTGTTTGTG, Bc3R: CACCTAGGTATTTCCTGGTA, Bc4F: CATCTTCTGGGAACGCACAT, Bc4R: ATCCACCCCCAAACGATTGT, Bc5F: CGTTTTCCAGCATTTCAAGT, Bc5R: CATCTCATATTCGTTCCTCA, Bc6F: ACTAGATTCGAGATTCAGTT, Bc6R: AAGGTGGTATGAGCGGTTTA, Bc7F: CCAGTTTCGAGGAGGTCCAC, Bc7R: GCCTTAGCGGATGTGAGGTA, Bc9F: CTCGTCATAACCACGCAGAT, Bc9R: GCAAGGTCTCGATGTCGATC, Bc10 F: TCCTCTTCCCTCCCATCAAC, Bc10R: GGATCTGCGTGGTTATGA). The primers were combined into three multiplex sets of primers 1, 5, and 10; 2, 3, and 6; and 4, 7, and 9. The forward primer were labeled with ROX (primer 1, 4), HEX (primer 7, 10), TAMRA (primer 2), and FAM (primer 3, 5, 6, and 9). The PCR program consisted of 30 cycles of denaturation, annealing, and elongation, with initial denaturation at 95 °C for 3 min, denaturation at 95 °C for 15 s, primer annealing at 60 °C for 30 s, elongation at 72 °C for 30 or 50 s, and final elongation at 72 °C for 3 min [48,49]. The analysis of the resulting fragments was performed with the 3130xl Genetic Analyzer (Applied Biosystems, Darmstadt, Germany) and GeneMapper 4.0. software. The polymer used was POP7TM (Thermo Fisher Scientific, Waltham, MA, USA; [48,49]).

### 2.2. Laccase Activity Assay

The inducement and determination of laccase activity was conducted in accordance with the methodology previously described by Umberath et al. [42]. A detailed methodology is given below.

#### 2.2.1. Inducement of Laccase Activity

For inducement, liquid cultivation in the presence of gallic acid (purity 98, Alfa Aesar, Ward Hill, MA, USA) was necessary [50]. 50 mL of medium containing 30 g*L 1 malt extract, 5 g*L 1 soy peptone, and 1 g*L 1 gallic acid was filled in a 250 mL Erlenmeyer flask in duplicate for each strain. The flasks were inoculated with $10^{5}$ spores and incubated at 25 °C for 3 days in the dark under continuous shaking. The mixture was homogenized using a T 25 digital Ultra-Turrax^®^ (IKA- Werke GmbH and CO. KG, Staufen, Germany), and 100 uL was transferred to MPA plates in triplicate (diameter 8.4 cm, filled with 10 mL MPA with a Dispensette^®^ S). The plates were incubated for another 11 days under the same conditions. A defined number of pieces (diameter 0.5 cm) were punched out and extracted with sodium acetate buffer (150 rpm at 35 °C for 90 min), and laccase activity was determined using the syringaldazine assay [51]. The absorption was measured with a FLUOstar Omega microplate reader spectrophotometer (BMG Labtech, Ortenberg, Germany) at 595 nm [52]. For quantification, a standard curve with laccase from *T. versicolor* was established.

#### 2.2.2. Determination of Laccase Activity

To determine the amount of biomass, a qPCR was carried out as described by Umberath et al. [42], as follows: Plates were lypholized and homogenized using liquid nitrogen. DNA was extracted using REDExtract N-Amp Plant PCR Kit (Merck KGaA, Darmstadt, Germany) using an adapted version of the primers described by Suarez et al. [53] and Diguta et al. [54] for quantification of *Botrytis cinerea* (Bc10nt_3R: 5-GGA GCA ACA ATT CGC ATT TCA AAC ATG CTG-3 and Bc10nt_3F: 5-GCT GTA ATT TCA ATG TGC AGA ATC CTG TCC CCG GT-3). The analysis was carried out using PowerTrack SYBR Green MasterMix (ThermoFisher Scientific, Vilnius, Lithuania) and ROX as a passive reference dye. The resulting DNA quantified was calculated using a 10-fold template solution of the respective strain and converted into an amount of DNA using an Epoch Microplate Spectrophotometer (BioTek, Winooski, VT, USA). The amount of DNA correlated to the biomass was then used in combination with the syringaldazine assay to obtain the corrected relative specific laccase activity. One unit (ULacc) was defined as the amount of enzyme oxidizing 1 µMol syringaldazine per minute. Correction of laccase activity by the corresponding biomass was conducted to obtain relative specific laccase activity [ULacc * mDNA 1]. The percentage of activity related to the water treatment was used as an indicator of inhibition. Negative values indicated induction of enzymatic activity. The laccase activity between the strains was compared by statistical analysis using *t*-tests.

### 2.3. Bioinformatics

Eight strains were further analyzed by next generation sequencing (NGS). The strains were selected among different regions, vintages, and grape varieties (see Table 1). For the NGS, the service from Eurofins (Eurofins Genomics) was used. The data were obtained using the Illumina NovaSeq technology, and the sequence mode NovaSeq PE 150 was chosen since it provides high-throughput analysis and high data output (Eurofins Diagnostics). The resulting data were in FASTAQ format and were used for downstream analysis. The following software was used for additional analysis: BCFtools, BWA, Cygwin, fastp, FastTree, GATK, JalView, Mafft, MEGA, Perl, Python, R, sambamba, samtools, Sentieon, snpEff, and VCFtools; see Table 2).

The NGS data were aligned to the known reference strain of *Botrytis cinerea* B05.10, which is available online at Ensembl (Assembly ASM83294v1, January 2015; [55], INSDC Assembly GCA 000143535.4, February 2015; [56]). The sample set was controlled for quality metrics as followed: total raw reads, total HQ reads, HQ bases(Q30), GC content, mean read length, and HQ reads in %.

#### 2.3.1. Mapping and Alignment

The mapping and alignment of the sequenced data were conducted using BWA and Sentieon (see Table 2). The mapping included the following statistics: total HQ reads, mapped reads, unique reads, deduplicated reads, mean coverage, and mean coverage (w/o duplicates). Alignments were visualized using JalView.

#### 2.3.2. Coverage Report and Distribution

For each sample, the mean coverage metrics were calculated. To provide fundamental data for the sample set total bases, mean coverage, uniformity, percentage of covered reference, and a summarized depth of coverage were analyzed.

#### 2.3.3. Variant Call Reports

For the variant call reports, SNP and InDel were called using Sentieon’s Haplotype-Caller (see Table 2) and applied with customized filters to filter false positive results using GATK Variant Filtration (see Table 2). The data underwent further analysis using special filters by Eurofins (Eurofins NDSC Food Testing GmbH, Hamburg, Germany). All data with a coverage depth of 30× or less and an allele depth less than 3× were subjected to a filter process utilizing the Eurofins High-Confidence Filter. Any data that did not meet the criteria for this filter were designated as FAILED and excluded from the analysis. Those data points matching the filter criteria were set to PASSED. However, for the purposes of general observation, a subset of the data was analyzed without the filter criteria. The detected variants were then annotated using snpEff (see Table 2).

#### 2.3.4. BLAST

Selected genes were blasted against the whole genome using BLAST+. The selected gene regions were blasted against the whole genome to find additional matches.

#### 2.3.5. Construction of Phylogenetic Tree

Selected gene regions based off of available data by Ensembl were used to construct the phylogenetic trees. The selected gene regions consisted of the laccase gene regions (Lac1, Lac2. Lac3, Lac4, Lac5, Lac6, Lac7, Lac8, Lac9, Lac10, Lac11, Lac12A, Lac12B, Lac13; see Table 3). The parameter settings for FasTree were as follows: “-gtr -nt -boot 1000”; gtr: generalized time-reversible model (GTR) was chosen because of its generalistic and neutral time-reversible model properties [57]; nt: the input data are based of nucleotide data; boot 1000: bootstrap calculation with 1000 iterations over the dataset. The phylogenetic trees were then visualized using MEGA software.

## 3. Results

Eight strains were selected for analysis using next generation sequencing (NGS). The resulting sequences were further analyzed and put into context with strain data.

### 3.1. Bioinformatic Results

The general sequencing data from the analysis were calculated using fastp to reduce poor-quality reads of the samples. The data are summarized in Table 4 and include the number of raw reads, HQ reads, HQ bases(30), GC content, and HQ reads.

The HQ reads were further analyzed using alignment tools. The alignment was conducted with reference to the *Botrytis cinerea* sequence from Ensembl accessible online at the European Nucleotide Archive (ENA, [56]). The results show that the majority of the sample data could be uniquely mapped to a specific region with a slightly less-mapped read of S1R1V4 (78.19%). The percentage of mapped reads for both S3R3V1 and S4R3V2 was relatively low (33.33%, 31.21%) in comparison to the other strains. After removal of the PCR duplicates (deduplicated reads), the number of reads remained high in all samples. The coverage exceeded 30× across the entire genome with the exception of S3R3V1 and S4R3V2 (see Table 5). To assess the mutation occurring in the strains, the variant metrics of SNPs and InDels were conducted for the whole genome (see Table 6) and for the laccase regions (see Table 7).

In addition to the variant metrics, the laccase gene regions were further analyzed for potential protein changes and other modifications. For potential impacts on the promoter and terminator regions of the related genes, 1500 bases downstream and 1500 bases upstream were also analyzed. There were several protein changes in the different strains compared to the reference, but the amount of highly impactful changes was relatively low. There were two regions in which a highly impactful change occurred: At chromosome NC_037318.1 position 1688641, the nucleotide C changed to A, resulting in a stop for S1R1V4, S6R4V4, S7R5V4, S8R5V4, and S2R2V4 except for S5R4V3, S3R3V1, and S4R3V2. At chromosome NC_037318.1 position 1691494, the nucleotide C changed to T, resulting in a stop for S1R1V4, S5R4V3, S8R5V4, and S2R2V4 expect for S6R4V4, S7R5V4, S3R3V1, and S4R3V2. The strain with the highest measured laccase activity, S6R4V4, was compared to all other tested strains. The amount of unique changes was relatively low, as indicated in Table 8. We further looked into a selection of genes potentially responsible for gene regulation, but again, there were no high impacts correlating with laccase activity.

#### Phylogenetic Tree

In total, 15 phylogenetic trees were calculated using MEGA. The phylogenetic trees consisted of all known laccase gene regions as well as a phylogenetic tree based on all laccase gene regions together. The tree for the combined data is provided below (see Figure 2), while the single gene region trees are provided in the Appendix A. For the phylogenetic tree of the combined regions, the tested samples were distinct with mostly high bootstrap values. The differences of the tested samples varied throughout the single gene region trees. Some regions showed no or low variety Appendix A, while others showed a high distinction (Appendix A), underlined by bootstrap values.

### 3.2. Laccase Activity

The laccase activity (LA) of the selected strains is shown below (see Table 9). The LA varied greatly between the tested strains, ranging from 57.53 mU/ng DNA to as little as 0.10 mU/ng DNA. The LA of the strain S1R1V4 was not measured but is included below to complete the dataset in comparison to the other results. Overall, the LA in strain S3R3V1, S4R3V2, and S5R4V3 was relatively low (0.1–0.5 mU/ng DNA), the LA of S2V2R4 and Ref a bit higher (6.0–8.2 mU/ng DNA), followed by increasing amounts of LA from S8R5V4 (14.94 mU/ng DNA), S7R5V4 (22.14 mU/ng DNA), and S6R4V4 (57.50 mU/ng DNA).

### 3.3. Strain Assessment

In total, 59 strains were assessed in the SSR-PCR. The raw data of this experiment, including the methodology, were previously published [48]. Here, the data were further analyzed to obtain a dataset which highlights the differences of the strains found in the SSR-PCR. The capillary sequencer fragment sizes were compared to each other using a heatmap. Since a total of nine SSR primer pairs were used, the amount of differences ranged from zero to nine. In addition, a heatmap containing only the selected and genome-sequenced strains is provided (see Figure 3).

A UPGMA tree was constructed based on the resulting fragment sizes (see Figure 4) showed distinct relationships for all the tested strains. There were two main groups, S1R1V4 and Ref, as well as the other strains. The other strains further subdivided into smaller groups.

## 4. Discussion

The results offer new insights into the genomic level of laccase genes in *Botrytis cinerea*. We compared eight strains from different regions and vintages with a reference strain. This was done in order to gain insight into the potential evolution of laccase as well as to obtain information on the laccase gene relationships. The results were compared to one another and then compared to their relative laccase activity (LA) and their genomic properties.Single-nucleotide polymorphisms (SNPs) and insertion–deletion mutations (InDels) were found in all strains with a relatively homogeneous distribution of variants across the laccase gene regions. The total number of changes ranged from 401 to 452, comprising 379–421 SNPs and 21–31 InDels. It is noteworthy that, despite having fewer mapped reads and therefore a lower number of SNPs and InDels in the total genome, the laccase regions of S3R3V1 and S4R3V2 exhibited the same number of SNPs and InDels as the strains with more mapped reads. This suggests that the laccase regions are relatively unique across the genome and can be aligned with a high degree of certainty. The strains with the most SNPs and InDels do not necessarily have a higher laccase activity, although the strain S6R4V4 had the most SNPs and InDels and the highest laccase activity, and the strain S5R4V3 had second highest SNPs, the same amount of InDels, and a low laccase activity. Both strains derived from the same location but different vintages. Therefore, SNPs and InDels might be related to regions as well. S7R5V4 and S8R5V4 had round about the same amount of SNPs as the strains S3R3V1 and S4R3V2 but a higher laccase activity. They derived from two different regions; however, the distance between the regions was only a few kilometers. This also indicates that SNPs might derive from regional aspects, but it alone cannot explain laccase activity.

Looking at the whole genomic level (Table A1) reveals a relatively homogeneous distribution of the variant effects across all strains, with the exception of strains S3R3V1 and S4R3V2, which derived from the vintages 2017 and 2020, respectively. Given that the SNPs and InDels mostly derive from the laccase regions, these variant effects are likely from those regions and can be compared to the other strains. Having a highly unique and conserved laccase region makes sense, since they are most important for overcoming plant defense mechanisms [18,58]. However, the results of those two samples have to be reviewed carefully, since the overall mapping quality was lower compared to the other samples. The amount of SNPs was higher than the amount of InDels, which is to be expected, since gene changes occur at a much higher frequency than InDels [59]. The number of changes was not correlated to the vintage or the region, indicating slower evolutionary changes in the laccase gene regions. In the time span of 6 years, no significant changes in SNPs or InDels could be observed. In general, SNPs and InDels are equally potent to cause altering effects on the genome and protein properties [59].

### 4.1. Conserved Laccase Regions

The phylogenetic tree of *Lac 4* and *Lac 5* (Appendix A) showed no difference between the tested strains. This could be due to a highly conserved region, exhibiting minimal to no change over time. The strain vintages range from 2017 to 2022, in which no clear divergence could be observed for these gene regions. It is possible that the divergence between some strains is recent, and therefore, the divergence time was insufficient to detect differences between the strains. Since the reference strain is even older than the tested samples, it is likely that these regions are highly conserved [60] and that LA cannot be explained by those gene regions. The patterns observed in *Lac 6*, *Lac 9*, and *Lac 12B* are similar, with little to no divergence (Appendix A). Notably, in the *Lac 6* and *Lac 9* region, the reference strain was distinctly separated from the majority of strains, indicating a gradual genomic shift in those regions. Conversely, the laccase regions *Lac 1*, *Lac 2*, *Lac 3*, *Lac 7*, *Lac 8*, *Lac 10*, *Lac 11*, *Lac 12A*, and *Lac 13* exhibited greater diversity, as evidenced by (Appendix A). *Lac 1*, *Lac 2*, and *Lac 3* have been previously analyzed in different aspects, for example, gene expression studies of *Lac 1* and *2* and *3* [61], studies based on the enzyme activity of *Lac 2* and *3* [62]. *Lac 7*, *Lac 8*, *Lac 11*, and *Lac 13* show an especially great diversity and have been relatively less studied. One study looked into *Lac 7* and *8* but focused on *Lac 2* and *Lac 3* [38]. Another study found no differences between three strains in the *Lac 2* and only some in the *Lac 7* region [62]. Conversely, we found great differences in both laccase regions.

### 4.2. Comparison of Strains

A subgroup of the strains S2R2V3, Ref, S6R4V4, and S5R4V3 was identified with a high degree of certainty (98%) for *Lac 12B*. The subgroup then dived further into smaller subgroups, but the bootstrap values were relatively low with 36% and 49% (see Appendix A). With regard to *Lac 9*, a subgroup comprising S2R2V4, S6R4V4, and S5R4V3 was identified, exhibiting a high degree of certainty (64%), (see Appendix A). In addition to the reference, both gene regions group the same strains together in a cluster, namely S2R2V4, S5R4V3, and S6R4V4. A similar pattern could be seen in *Lac 2*, *Lac 7*, *Lac 10*, *Lac 12A*, and *Lac 13*. The relation between S2R2V4 and the reference strain (Ref) seems to be close in different genomic regions, being in the same groups in *Lac 3*, *Lac 6*, *Lac 10*, *Lac 11*, and *Lac 12B*. Other strains which grouped together in most laccase gene regions were S7R5V4 and S8R5V4 as well as S3R3V1 and S4R3V2.

### 4.3. Regionality

As already mentioned, a similar relationship can be observed for S7R5V4 and S8R5V4. The strains were sampled from the same location and time, situated only two meters apart from each other. Diversity of *Botrytis* on a regional level is likely to be present [63]. Interestingly, they showed different LA (22.14 and 14.94) despite being grouped together or relatively close in all laccase Regions. The involvement of numerous additional proteins in LA, such as BcnoxA and BcnoxB [64], underscores the limitation of assessing LA differences based solely on laccase gene regions. Nevertheless, these observations provide valuable insights. The strain S2R2V4 is from a distant region in France but still shows similarities in its respective phylogeny even though the climate between both regions is different. While the Sauternes in the mid-west of France has a maritime climate [65], the weather in mid-west Germany is rather dry [66]. The Sauternes is a unique wine-growing region featuring special climate conditions. During the harvest season, high humidity favors the production of *Botrytis cinerea*, which will then dissipate during warmer middays and in turn favor the noble rot [44,67]. The different climate conditions may favor different *Botrytis* strains. It is important to note that there are no fundamental differences between noble rot strains and grey mold or sour rot strains, and the quality of noble rot is dependent on environmental conditions during the harvest period [44]. The similarities between the strains on a genomic level could be due to regional differences based on climatic conditions. The S2R2V4 strain is often grouped with the reference strain, which is originated from California, again providing a potential relationship between regionality and strain differentiation of the laccase regions. When comparing the LA, the strain is in fact closer to the LA of the reference strain, but this could be a sampling bias; as previously mentioned, this pattern does not continue throughout all tested samples. The laccase activity does not seem to correlate with the phylogenetic trees of the laccase regions. Rather, the main distribution is of regional factors, likely influenced by environmental conditions. For example, the availability of free surface water and a relatively high humidity of 90% allow for an optimal infection of the plant host [68]. Therefore, high laccase activity might be correlated with low humidity and free surface water. However, the temperature also plays an important role. An optimal infection is achieved in combination with 15–20 °C [68]. Therefore, regions such as the Diedesfeld, Edenkoben, and Heppenheim, which are more wet, do met the requirements for high humidity and free surface are water and should have higher laccase activity. While this is partly true for the Edenkoben and Heppenheim strains, the Diedesfeld strains did not have high laccase activity.

The *Lac 11* region provides a relatively clear distinction of all regions samples, putting all regions in groups, although for R 3 (Region 3-Diedesfeld, Germany) and R 4 (Region 4-Heppenheim, Germany), no significant bootstrap value were calculated to clearly distinguish those two groups. This makes sense, since both regions are only about 60 km apart, whereas the other regions are more distant. *Lac 7* and *Lac 8* were also able to distinguish most of the strains by regionality. These findings are supported by numerous studies which found a high genetic diversity of *Botrytis cinerea* in different regions and climates, as reviewed by [69].

### 4.4. Vintages

The strains S5R4V3 and S6R4V4 were collected from the same region but from two different vintages. The measured LA varied strongly between them (0.481 in 2021 and 57.53 in 2022). Different weather conditions can favor certain strains by selection [68]. Once more, the two strains were found to be grouped together in the laccase gene regions, which indicates a close relationship between them. The large difference in LA might be due to specific SNPs or InDels in those genomic regions, since small changes can have significant effects on the mRNA produced [70]. Strains S3R3V1 and S4R3V2 were sampled in another region about 3 years apart and showed roughly the same amount of LA. Those strains seem to be closely related, being grouped together in most of the laccase gene regions and in the SSR-PCR as well. The close relationship indicates a slow strain adaption in this specific region. There was no laccase gene region which grouped the tested strains by their vintage alone. This could be due to the complexity of different *Botrytis* strains, which are more likely to adapt to environmental factors which are not necessarily related to vintage but could also be due to a low sample amount. Analyzing local strain changes and evolution could reveal more differences than a broad analysis over multiple regions, especially since climate change-related factors are more homogeneous in one region than in multiple regions at once. The strain S5R4V3 had a very low laccase activity compared to S6R4V4. S5R4V3 was from the 2021 vintage with high humidity and rain periods during the late season of harvest, while the 2022 vintage showed lower amounts of rain periods [71]. This could show that evolutionary change is low in certain laccase gene regions but that some regions show a faster change rate in the tested samples. Having conserved laccase genes serves as an important tool in pathogenity. These findings are especially important for long-term strategic planing related to strain-dependent treatments of grapes and adaption to pathogen severity in the vineyard. Genomic shifts and alteration in strains regarding LA can be quick, changing from vintage to vintage depending on weather conditions and other factors such as competition or favorable selection of strains, or they can be slow and resilient to change.

### 4.5. Is There a Predictor for Laccase Activity?

The first consideration of a predictor for laccase activity was the high-impact changes in the laccase gene regions. The stop gained in *Lac 3* at 1688641 is in fact correlated with the LA of the strains. All strains that gained a stop codon had higher measured LA then the strains which did not. However, this change cannot explain the difference in LA between the strains. For example, the LA of S6R4V4 was way higher than the other strains with high LA. For S1R1V4, no LA was measured. Therefore, the findings have to be considered as a new insight rather than a certain result, but they seem to be an indicator for higher LA. The stop at position 1691494 was about 1400 bases upstream of the *Lac 3* gene. Here, no clear correlation between LA could be observed. It is possible that this stop is not involved in the termination of the *Lac 3* gene or is irrelevant for the change of the coding protein. Apart from this change, the most changes occurred in the genome rather than in the terminator or promoter region. Taking all strains into account, there was no clear observation on specific protein changes responsible for a higher or lower laccase activity. It is possible, that there are relevant protein changes responsible for a LA change, however due to the complexity of the genomic level, no clear correlation could be detected. The matter is even more complicated by the involvement of regulatory proteins [30,64,72]. There are several proteins involved in the regulation of laccase and, therefore, potentially LA. We looked into a selection of those proteins important for *Botrytis* virulence, namely BcNoxA, responsible for development of sclerotia [64]; BcNoxB, responsible for penetration [64]; and BcPLS1, related to appressorium penetration [30]. However, no concise evidence for protein changes specific to high or low LA could be found. Again, no significant changes between strains with high LA and low LA could be observed. The difference in LA might be due to a combination of different protein changes in the genomic regions as well as in the regulatory gene region and the promoter and terminator regions. Based on the low sample size and the order of complexity, we cannot pinpoint any specific genomic changes responsible for the LA. The results show the complexity of LA correlation between different strains and in general. To assess the LA, gene expression needs to be considered. Some studies already showed methods for the gene expression analysis of specific laccase regions [61].

### 4.6. SSR-PCR as a Tool for Laccase Gene Analysis

The SSR-PCR results transformed into an UPGMA tree showed an absence of a clear correlation between the laccase gene regions and the SSR-PCR strain differentiation. While the SSR-PCR was based on SSRs from conserved regions [43], the phylogenetic trees of the laccase regions were based on their respective laccase gene region. Consequently, when analyzing laccase genes of different strains it is not possible to obtain a distinct result by phylogenetic analysis alone. It is necessary to conduct a strain differentiation in addition to further analysis. However, it might be possible to get a relatively precise oversight on the data when analyzing only specific laccase regions such as *Lac 1*, *Lac 7*, *Lac 8*, *Lac 11*, and *Lac 13*. Those specific regions were able to distinguish the strains with a high certainty. With regard to the study of alterations in the laccase gene of the *Botrytis* genome, the aforementioned regions may serve as a suitable point of departure, given their relatively low degree of conservation. Conversely, in the context of gene expression studies or mutation analysis, the conserved laccase regions may warrant further investigation, as they appear to remain largely unaltered over time or in response to regional climatic variations. Analysis would add on generic differentiation of *Botrytis* strains, such as multilocus sequence typing (MLST, [73]).

## 5. Conclusions

Our initial hypothesis could only be partially confirmed, as there was no clear correlation of SNPs or InDels in the open reading frame or regulatory level. The combined results of SSR-PCR and strain differentiation as well as laccase gene analysis of *Botrytis cinerea* is complex and difficult to understand. Many uncertainties need to be discussed and interpreted in the correct context. Here, we have shown that looking at just a few aspects of *Botrytis* at the genomic level is not sufficient to obtain coherent data. Instead, the whole context from the genomic level as well as direct strain analysis and laccase activity detection need to be conducted to understand the correlation between different strains. We have also been able to show that strain evolution is indeed different between regions and years and that changes in laccase activity do not seem to be related to specific genes but are fundamentally dependent on SNPs or InDels mutations. The main factor of change seems to be regional factors, namely climatic and environmental conditions. Some less-researched laccase genes, for example *Lac 11* and *Lac 13*, provide insightful information on the assessment of *Botrytis cinerea*. Other regions, such as *Lac 4* or *Lac 5*, seem to be highly conserved and resistant to change, serving as potential markers for strain analysis studies. It its important to keep up to date with the ongoing changes in the pathogen landscape, as changes can be rapid and devastating. New methods need to be established, not only for prevention and curation, but also for understanding the adaptability and changes. Future research could look further into genomic relationships, possibly including more relevant regions such as gene regulatory proteins or expanding on less-studied gene regions such as *Lac 7*, *Lac 8*, *Lac 11*, or *Lac 13*. More strains could be placed in context to assess more information on genomic changes. On another note, gene expression studies are equally important and could also benefit from genomic insights. The results indicate that gene expression studies could provide highly important insights into this topic and clarify our hypothesis. When diving deeper into the matter, looking at different grape varieties, especially pathogen-resistant varieties, could provide even more insight into *Botrytis*-related problems. Research in this field can benefit viticulture and oenology on different levels, for example, in preparation during harvest or onset of the pathogen’s severity. Understanding the regionality of strains will help researchers to find recommendations and solutions for winegrowers as well as to stay on par with climate change-related regional changes.

## Figures and Tables

**Figure 1 microorganisms-13-00483-f001:**
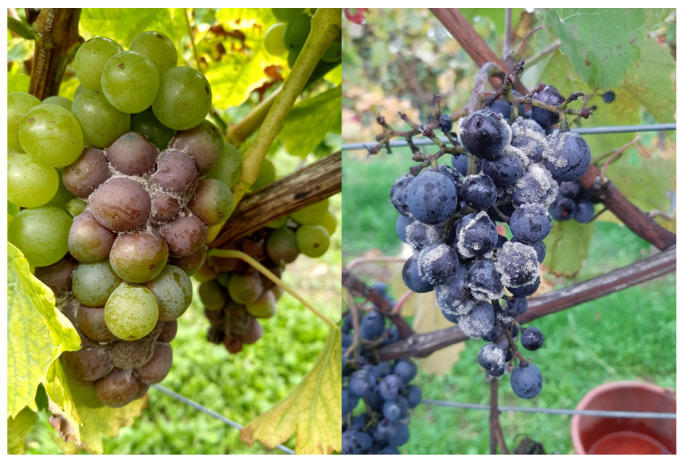
Sample Pictures of visible sporulating grape bunches from the collection sites. The grape bunches were transferred to the laboratory in bags and cultivated on malt agarose plates.

**Figure 2 microorganisms-13-00483-f002:**
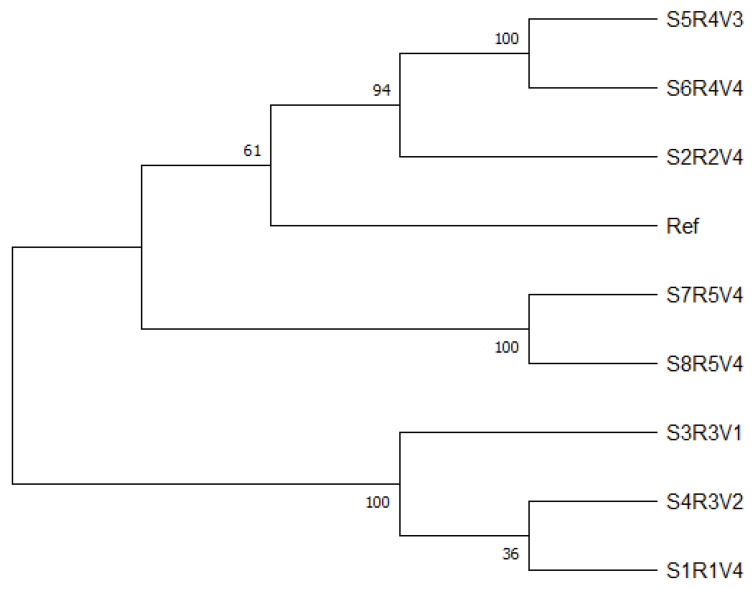
Maximum likelihood tree of all laccase gene sequences of *Botrytis cinerea* compared for every strain. The strains selected were from different regions and vintages. Bootstrap values were calculated from 1000 iterations and are displayed in %.

**Figure 3 microorganisms-13-00483-f003:**
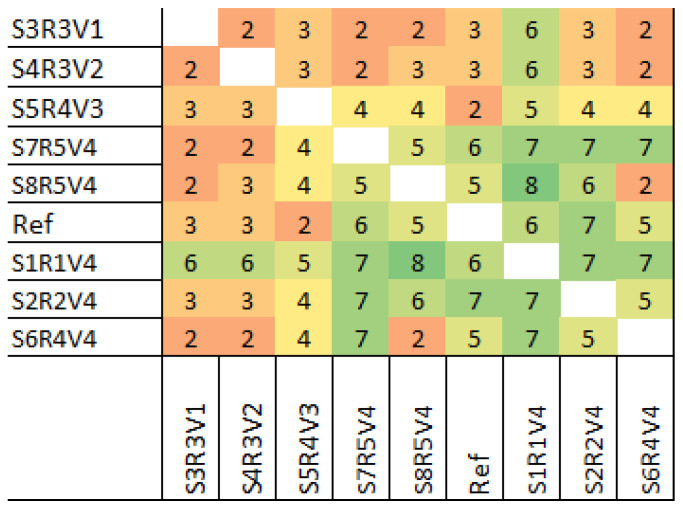
Results of the SSR-PCR analysis. The total differences of all primer pairs used in the SSR-PCR were used to generate a heatmap, as previously published [48], to show their differences throughout the strains. A minimum of zero (red) to nine (green) was possible.

**Figure 4 microorganisms-13-00483-f004:**
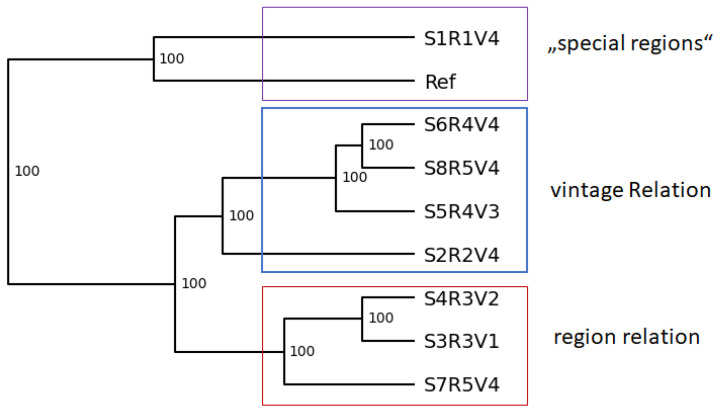
UPGMA Tree of the selected *Botrytis cinerea* strains based on the results of the SSR-PCR. Bootstrap of 1000 was calculated and is indicated on the branches in %. Potential group relations are highlighted (purple: untypical vintage regions–reference strain and strain from a non-viticulture-related urban region; blue: close relation in vintage; red: close regional relation; Palatinate region).

**Table 1 microorganisms-13-00483-t001:** Sampled and sequenced strains. The samples are given a namecode based on the strain (S), the region (R), and the vintage (V).

Strain	Region	Vintage	Namecode
Strain 1	Bonn	2022	S1R1V4
Strain 2	Bordeaux	2022	S2R2V4
Strain 3	Diedesfeld	2017	S3R3V1
Strain 4	Diedesfeld	2020	S4R3V2
Strain 5	Heppenheim	2021	S5R4V3
Strain 6	Heppenheim	2022	S6R4V4
Strain 7	Edenkoben	2022	S7R5V4
Strain 8	Edenkoben	2022	S8R5V4

**Table 2 microorganisms-13-00483-t002:** Software and tools used for data analysis and data acquisition.

Software	Version	Description
BCFtools	1.21 ^1^	BCFTools is a utility tool used to process and analyze data in the BCF (binary call format) and VCF (variant call format) formats.
BWA	0.7.17	BWA is designed for mapping reads against a large genome, especially for low-divergent sequences.
BLAST+	2.16.0+	BLAST+ is a tool from NCBI used to find sequence targets in the genome.
Cygwin	3.5.4	Cygwin provides a Linux-based environment under Windows to use bioinformatic tools.
fastp	0.20.0	Fastp is used for the quality control and processing of FastaQ files.
FastTree	2.1.11	FastTree provides a fast construction of phylogenetic trees, and is optimized for big datasets.
GATK	3.7	GATK is a general toolkit for SAM/BAM/VCF files.
JalView	2.11.4.1	JalView is an interactive tool used to visualize and analyze sequence alignments, as well as editing.
Mafft	v7.490	Mafft is a software package for the sequence alignment of datasets.
MEGA	11.0.13	MEGA is a software tool used to visualize and analyze phylogenetic trees.
Perl	5.28.2	Perl is a high-level, general-purpose programming language.
Python	3.9.16	Python is an universal programming language, supporting the construction of bioinformatic tools.
R	4.1.3	R is a programming language focused on statistical analysis and computing.
sambamba	0.6.8	Sambamba is an analysis tool for the processing of SAM and BAM files as well as sequence alignments.
samtools	1.20 ^2^	Samtools is able to manipulate various data in the SAM format.
Sentieon	202308	Sentieon is a bioinformatic software used to speed up GATK pipelines. It is used for variant detection and genome analysis.
snpEff	4.3	snpEff is a tool for the annotation and interpretation of genetic variants.
VCFtools	0.1.17 ^3^	VCFtools is a program package for various analyses of VCF files.

An older version was used throughout analysis: ^1^ 1.15, ^2^ 1.10, ^3^ 0.1.16.

**Table 3 microorganisms-13-00483-t003:** Gene regions used for construction of the phylogenetic trees.

Name	GeneID	Chromosome	Position START	Position END
Bclcc1	Bcin01g07950	1	2771031	2774287
Bclcc2	Bcin14g02510	14	987192	989737
Bclcc3	Bcin09g04830	9	1687129	1690028
Bclcc4	Bcin05g03550	5	1286300	1288676
Bclcc5	Bcin01g07190	1	2506945	2509587
Bclcc6	Bcin15g03330	15	1163985	1167472
Bclcc7	Bcin02g07640	2	2716773	2719282
Bclcc8	Bcin01g00800	1	304243	307216
Bclcc9	Bcin07g06780	7	2512424	2514895
Bclcc10	Bcin09g02050	9	756270	759140
Bclcc11	Bcin08g00050	8	18476	19138
Bclcc12A	Bcin09g04420	9	1549603	1554714
Bclcc12B	Bcin13g00110	13	45322	47416
Bclcc13	Bcin02g02780	2	1014404	1016983

**Table 4 microorganisms-13-00483-t004:** Sequence data of the analysis including raw reads and quality of reads.

Strain	Total Raw Reads	Total HQ Reads	HQ Bases (Q30)	GC Content	HQ Reads %
S1R1V4	19.98 M	19.68 M	94.95%	39.62%	98.43%
S2R2V4	13.94 M	13.1 M	94.13%	42.83%	98.26%
S3R3V1	16.97 M	16.65 M	94.43%	41.76%	98.09%
S4R3V2	9.5 M	9.29 M	93.26%	49.37%	97.82%
S5R4V3	12.59 M	12.41 M	94.71%	42.93%	98.52%
S6R4V4	11.78 M	11.63 M	94.06%	42.57%	98.33%
S7R5V4	12.8 M	12.55 M	93.95%	42.72%	98.06%
S8R5V4	14.26 M	13.95 M	93.07%	42.99%	97.84%

**Table 5 microorganisms-13-00483-t005:** Alignment statistics. Mapping metrics per sample.

Strain	Total HQ Reads	Mapped Reads	Unique Reads	Deduplicated Reads	Mean Coverage (w/o Duplicates)
S1R1V4	19.68 M	15.39 M (78.19%)	14.99 M (76.16%)	13.04 M (87.00%)	44.54×
S2R2V4	13.10 M	12.34 M (94.14%)	11.91 M (90.89%)	9.83 M (82.54%)	33.86×
S3R3V1	16.65 M	5.55 M (33.33%)	5.27 M (31.67%)	4.61 M (87.51%)	14.99×
S4R3V2	9.29 M	2.90 M (31.21%)	2.77 M (29.77%)	2.41 M (87.23%)	7.96×
S5R4V3	12.41 M	11.26 M (90.72%)	10.91 M (87.97%)	9.57 M (87.71%)	32.90×
S6R4V4	11.63 M	10.93 M (93.92%)	10.59 M (91.01%)	9.12 M (86.14%)	31.31×
S7R5V4	12.55 M	11.92 M (95.01%)	11.51 M (91.77%)	9.39 M (81.56%)	32.44×
S8R5V4	13.95 M	13.03 M (93.36%)	12.54 M (89.84%)	8.83 M (70.46%)	30.58×

**Table 6 microorganisms-13-00483-t006:** Variant metrics per sample based on the whole genome. The sample name, total variants, SNPs, and Indels are listed.

Strain	Total	SNP	InDel
S1R1V4	210,020	190,776	19,244
S2V2R4	139,779	129,794	9985
S3R3V1	1970	1823	147
S4R3V2	839	782	57
S5R4V3	138,219	127,992	10,227
S6R4V4	113,218	105,636	7582
S7R5V4	128,434	119,012	9422
S8R5V4	105,804	98,347	7457

**Table 7 microorganisms-13-00483-t007:** Variant metrics filtered for the laccase regions (see Table 3). The sample name, total variants, SNPs, and Indels are listed.

Strain	Total	SNP	InDel
S1R1V4	409	382	27
S2V2R4	441	413	28
S3R3V1	409	382	27
S4R3V2	421	397	24
S5R4V3	446	415	31
S6R4V4	452	421	31
S7R5V4	401	380	21
S8R5V4	403	379	24

**Table 8 microorganisms-13-00483-t008:** Unique gene variations for S6R4V4. The strain had the highest measured laccase activity. In the table below are all unique changes for S6R4V4 in the gene regions of all laccase genes, excluding the low-mapped read results of S3R3V1 and S4R3V2.

CHROMOSOME	POSITION	REF BASE	OBS BASE	Bonn	HP2021	HPP	PM21	PM32	Bordeaux
NC_037311.1	2718688	G	A	p.Ser572Asn	p.Ser572Asn	-	p.Ser572Asn	p.Ser572Asn	p.Ser572Asn
NC_037318.1	755110	T	C	p.Ser777Ser	p.Ser777Ser	-	p.Ser777Ser	p.Ser777Ser	p.Ser777Ser
NC_037318.1	755245	G	A	p.Ser822Ser	p.Ser822Ser	-	p.Ser822Ser	p.Ser822Ser	p.Ser822Ser
NC_037318.1	1685956	T	C	p.Thr64Thr	p.Thr64Thr	-	p.Thr64Thr	p.Thr64Thr	p.Thr64Thr

**Table 9 microorganisms-13-00483-t009:** Measured laccase activity (LA) for each strain.

Strain	Laccase Activity [mU/ng DNA]	SD
S1R1V4	n.d.	n.d.
S2R4V2	8.19	0.05
S3R3V1	0.10	0.02
S4R3V2	0.15	0.14
S5R4V3	0.48	0.07
S6R4V4	57.50	3.18
S7R5V4	22.14	2.42
S8R5V4	14.94	0.02
Ref	6.16	1.53

## Data Availability

The original contributions presented in the study are included in the article/Appendix A; further inquiries can be directed to the corresponding author.

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
