# Peer review of "Putting Laccase Gene Differences on Genomic Level into Context: An Analysis of Botrytis cinerea Strains from Grapes"

_microorganisms, 2025, doi:10.3390/microorganisms13030483_

Round 1

Reviewer 1 Report

Comments and Suggestions for Authors

The manuscript, entitled "Putting Laccase Gene Differences on Genomic Level into Context: An Analysis of Botrytis cinerea Strains from Grapes", examines various aspects of laccase enzyme production in a globally important pathogenic fungus.
The introduction is sufficiently detailed. The materials and methods section presents the applied procedures, materials and processing methods.
The presentation of the results is well illustrated and illustrative.
The evaluation and discussion of the results are sufficiently detailed, analyzing a sufficient number of international references and citations.
Gene expression studies are important and could also benefit from genomic insights. The results indicate that these types of studies could provide highly important basic knowledge for this topic. Their findings offer new insights into the underlying mechanisms through which B. cinerea infects its hosts.
I recommend publishing the manuscript as a scientific article.

Author Response

Dear Reviewer,

Kind regards

The authors

Reviewer 2 Report

Comments and Suggestions for Authors

The study, "Putting Laccase Gene Differences on Genomic Level into Context: An Analysis of Botrytis cinerea Strains from Grapes," analyses the genomic differences in laccase genes between strains of the bacterium that are native to different regions and vintages. This work is significant because it investigates the evolutionary and regional diversity of these genes and how they might affect pathogen virulence and laccase activity. Even if this work provides valuable findings, several parts need to be rearranged, and improved for its readability and scientific validity.

1.       An overview of Botrytis cinerea and its importance in viticulture is given in the introduction. It's unclear why laccase genes should be the focus of attention. Explain the rationale behind the selection of laccase genes as the primary focus of this genomic investigation.

2.       The sample process is clearly described in the section sample and Strain Differentiation (Lines 73–82), although there are no specifics regarding replicates or controls. Specify whether controls were used and the number of replicates used for each strain.

3.       The methods used for assessing laccase activity are described in length in the section Laccase Activity Assay (Lines 83–118), but no mention is made of the statistical analysis that was used to compare activity between strains. Indicate which statistical techniques (such as ANOVA and t-tests) were applied to the analysis of the laccase activity data.

4.       A detailed list of the software and tools is provided, it is unclear what criteria were used to choose high-confidence variants. Give further information about the filters or criteria that are used to filter out low-quality data. Although phylogenetic trees are a suitable tool, the Generalized Time Reversible (GTR) model's rationale is absent.

5.       It is suggested to provide a brief explanation of why the GTR model was chosen.

6.       Although the results are structured, there is little investigation of the relationship between laccase activity and genetic variations. Talk about the possible effects of particular genomic changes (such as SNPs and Indels) on laccase activity and why they were more common in particular strains.

7.       The discussion offers helpful details, however, there are still some speculative aspects, especially when it comes to the evolutionary importance of conserved laccase regions.

8.       It is suggested to provide more sources to back up these claims, or classify them as hypotheses. Although intriguing, nothing is known about the relationship between laccase activity and regionality. Describe the potential effects of environmental factors on the observed regional variations in genomics variation.

9.       Some of the figures are lacking a descriptive legend. Make sure that every figure has thorough legends that describe the information and its significance. Although helpful, the UPGMA tree (Figure 3) might use some annotation or color-coding to draw attention to important results. Indicate regional or vintage groupings with annotations or color codes.

10.   Make sure that all of the references are relevant, up to date, and formatted consistently with journal titles and DOI links following the publication's guidelines.

11.   On line 7, "the laccase genes seem to be correlated with the regionality..." – Address how it affects our understanding of pathogen adaptation in generally.

12.   On line 30, "The high virulence factors of Botrytis are primarily regulated by the production and expression of laccase..." – Explain the relationship between plant defence mechanisms and laccase activity.

13.   On line 121, it says "The data was obtained using the Illumina NovaSeq technology..." – Explain why this sequencing platform was chosen over others.

14.   On line 202, it says "The differences of the tested samples varied throughout the single gene region trees..." – Name the gene regions that exhibited the most variation.

15.   On line 266, "While Lac 1, Lac 2 and Lac 3 have been previously analyzed..." Provide specific studies that has investigated these areas.

Author Response

Dear Reviewer,

Kind regards

The authors

Reviewer 3 Report

Comments and Suggestions for Authors

Pentru autori:

Dear authors,

The manuscript needs clarification. It is interesting from the point of view of the topic but complicated to understand exactly what the practical applicability is and the methodology needs major clarification.

Here are some suggestions for improvement and requests for clarification:

Line 74: Related to: <The strains were collected primarily... >

Please be more specific : Probably something like: The strains of B. cinerea

Line 74-75: Related to <All strains were isolated from visible sporulating grape berries ...>

A few images of the collection support material, that is, grape berries, would facilitate understanding of the working method.

Lines 78-79: Related to <A list of the 78 exact locations of the strains was previously published>

You should still provide the locations even if you have done studies in them and have already made them public. Simply provide them in another form (via satellite image) because you cannot refer to another publication. From what I found, it is another article (in Microbiology Research). Provide the reader with all the data in this manuscript.

In addition, the objective of this study is <to highlight strain differences in specific regions that may be responsible for infection efficiency.> So, a detailed description of these locations is necessary.

Lines 81-82: Related to <The strains were differentiated by SSR-PCR as previously described ([40]). >

Same situation as the previous one. It is necessary to describe the method(s), even if you have done so in an already published article, since here we are talking about another journal, another manuscript and another objective. In my opinion, this kind of approach/expression (<...as previously describe... >) is done when, in the same manuscript, you refer to the aspect in question.

Lines 84-85: Related to <The inducement and determination of laccase activity was condcuted in accordance with the methodology previously described by Umberath et al. >

Please detail the methodology, for all readers and for a better understanding. These references from one article to another do not ensure the fluency and connection of the steps in the methodology.

Table 1: Pay attention to the collection period because for Strain 3 and 4 the collection year appears 2017 and 2020 respectively, or in the text, above (lines 74-75 ) you mention that <...were collected primarily during the 2022 vintage, with additional sampling conducted during the 2021 vintage. > Please be more clear. What is actually the collection period/years?

Conclusions: Please justify the practical applicability of the study and here I refer to the certainty of the results obtained because in the objectives you say that they play a role in: <...the development of new strategies in viticulture and oenology... >

Kind regards,

R

Author Response

Dear Reviewer,

Kind regards

The authors

Reviewer 4 Report

Comments and Suggestions for Authors

The authors of the manuscript submitted to Microorganisms describe the results of the analysis of Botrytis cinerea strains in grapes.

The authors combine the well-known, but uninformative methods of SSR-PCR and high-throughput NGS (just for eight samples). It cannot be said that the obtained results were surprising or outstanding, in that case, the authors would probably have chosen another journal.

Nevertheless, the article seems competently written, the experiments were carefully conducted, and the interpretation of the results does not raise doubts. The reviewer suggests adding a hypothesis formulation to the introduction and whether it was proven or refuted in the Conclusions. Reviewer believes that this will be useful for the authors to understand the goal-setting, and for the reader to better understand the motivation of the authors

Also, the Introduction section should include a description of why the authors chose certain research methods.

Minor issues

1) Eight strains were selected for analysis using Next Generation Sequencing / how were the strains selected?

2) Figures A1-A14 can be moved to Supplementary

After the revision, the article might be recommended for publication.

Author Response

Dear Reviewer,

Kind regards

The authors

Round 2

Reviewer 3 Report

Comments and Suggestions for Authors

Dear Authors,

Your manuscript has been revised according to your suggestions and the answers have convinced me enough.

Personally, I am satisfied with this new version.

Kind regards,

R

Author Response

Dear Reviewer,

Your comment: Your manuscript has been revised according to your suggestions and the answers have convinced me enough.

Answer: Thank you for your positive feedback.

Kind regards

Reviewer 4 Report

Comments and Suggestions for Authors

The manuscript might be recommended for publication

Author Response

Dear Reviewer,

please see attachment.

Kind regards
